# Transcript Markers from Urinary Extracellular Vesicles for Predicting Risk Reclassification of Prostate Cancer Patients on Active Surveillance

**DOI:** 10.3390/cancers16132453

**Published:** 2024-07-04

**Authors:** Kati Erdmann, Florian Distler, Sebastian Gräfe, Jeremy Kwe, Holger H. H. Erb, Susanne Fuessel, Sascha Pahernik, Christian Thomas, Angelika Borkowetz

**Affiliations:** 1Department of Urology, Faculty of Medicine, University Hospital Carl Gustav Carus, Technische Universität Dresden, 01307 Dresden, Germany; kati.erdmann@ukdd.de (K.E.); sebastian.graefe@ukdd.de (S.G.); jeremykosasih.kwe@ukdd.de (J.K.); holger.erb@ukdd.de (H.H.H.E.); christian.thomas@ukdd.de (C.T.); angelika.borkowetz@ukdd.de (A.B.); 2National Center for Tumor Diseases (NCT), German Cancer Research Center (DKFZ), Faculty of Medicine and University Hospital Carl Gustav Carus, Technische Universität Dresden, Helmholtz-Zentrum Dresden-Rossendorf (HZDR), 01307 Dresden, Germany; 3German Cancer Consortium (DKTK), Partner Site Dresden, 01307 Dresden, Germany and German Cancer Research Center (DKFZ), 69120 Heidelberg, Germany; 4Department of Urology, Nuremberg General Hospital, Paracelsus Medical University, 90419 Nuremberg, Germany; florian.distler@klinikum-nuernberg.de (F.D.); sascha.pahernik@klinikum-nuernberg.de (S.P.)

**Keywords:** active surveillance, biomarker, liquid biopsy, monitoring, prediction, prostate cancer, quantitative PCR, risk reclassification, transcripts, urinary extracellular vesicles

## Abstract

**Simple Summary:**

Active surveillance is the preferred treatment strategy for low-risk prostate cancer and includes regular monitoring by control biopsies, which bear the risk of various side effects. In case of risk reclassification, therapy is switched to radical treatment. Currently used clinical parameters only possess a limited capability to indicate risk reclassification. Molecular markers identified via liquid biopsies, such as urine, could facilitate the detection of aggressive disease since they provide a more global assessment of prostate cancer than tissue biopsies. Moreover, an improved predictability could reduce the number of control biopsies needed during active surveillance. In this study, we identified a set of molecular markers from the urine of men on active surveillance that could predict the outcome of control biopsies. The combination of these molecular markers with clinical parameters resulted in further improved predictability of risk reclassification and, thus, has the potential to refine the monitoring strategies in active surveillance.

**Abstract:**

Serum prostate-specific antigen (PSA), its derivatives, and magnetic resonance tomography (MRI) lack sufficient specificity and sensitivity for the prediction of risk reclassification of prostate cancer (PCa) patients on active surveillance (AS). We investigated selected transcripts in urinary extracellular vesicles (uEV) from PCa patients on AS to predict PCa risk reclassification (defined by ISUP 1 with PSA > 10 ng/mL or ISUP 2-5 with any PSA level) in control biopsy. Before the control biopsy, urine samples were prospectively collected from 72 patients, of whom 43% were reclassified during AS. Following RNA isolation from uEV, multiplexed reverse transcription, and pre-amplification, 29 PCa-associated transcripts were quantified by quantitative PCR. The predictive ability of the transcripts to indicate PCa risk reclassification was assessed by receiver operating characteristic (ROC) curve analyses via calculation of the area under the curve (AUC) and was then compared to clinical parameters followed by multivariate regression analysis. ROC curve analyses revealed a predictive potential for AMACR, HPN, MALAT1, PCA3, and PCAT29 (AUC = 0.614–0.655, *p* < 0.1). PSA, PSA density, PSA velocity, and MRI maxPI-RADS showed AUC values of 0.681–0.747 (*p* < 0.05), with accuracies for indicating a PCa risk reclassification of 64–68%. A model including AMACR, MALAT1, PCAT29, PSA density, and MRI maxPI-RADS resulted in an AUC of 0.867 (*p* < 0.001) with a sensitivity, specificity, and accuracy of 87%, 83%, and 85%, respectively, thus surpassing the predictive power of the individual markers. These findings highlight the potential of uEV transcripts in combination with clinical parameters as monitoring markers during the AS of PCa.

## 1. Introduction

Prostate cancer (PCa) is the second-most-common cancer and the fifth-leading cause of cancer-related death in men worldwide [1]. However, low- and early intermediate-risk PCa is associated with a low probability of metastasizing and cancer-specific mortality [2,3]. Therefore, these PCa can be primarily managed by active surveillance (AS). Although the selection criteria for AS slightly vary between the current national and international guidelines, they are primarily based on the serum level of the prostate-specific antigen (PSA), the initial tumor grading (Gleason Score) according to the International Society of Urological Pathology (ISUP), the number of positive biopsy cores, and the local staging of the tumor [4]. Some protocols, such as the PRIAS project (Prostate Research International: Active Surveillance), additionally consider a PSA density (ratio of serum PSA and prostate volume; PSAD) of ≤0.2 ng/mL^2^, among others, as selection criteria for AS enrollment [5].

AS allows the patient to defer definitive treatment while regularly monitoring the tumor growth. In the event of tumor progression or risk reclassification into a more aggressive grading, a definitive treatment, such as prostatectomy or radiotherapy, is carried out. Hence, AS can delay therapy-associated complications and side effects, such as incontinence and impotence in the case of radical prostatectomy or chronic cystitis, chronic proctitis, or secondary malignancies in the case of radiotherapy. Consequently, AS maintains men’s quality of life for a longer time while showing the same long-term overall and cancer-specific survival rates as radical treatment for patients with low- and early intermediate-risk PCa as presented in the ProtecT trial [6]. However, AS patients had a doubled risk of developing metastasis, and six out of 10 patients converted to radical treatment during the clinical course [6]. Furthermore, the misclassification at AS enrollment and, thus, a delay of radical treatment might be associated with losing the window of curability. Therefore, regular monitoring of AS patients is essential.

Contemporary AS protocols recommend regular PSA monitoring, magnetic resonance imaging (MRI) of the prostate, and repeat biopsies to identify early signs of tumor progression and risk reclassification [4]. However, biopsies are associated with complications like infections or bleeding. In order to avoid invasive control biopsies and to predict the risk of tumor reclassification, it is absolutely necessary to identify clinical or molecular biomarkers. It has been demonstrated that an initial MRI was associated with a lower risk of tumor reclassification at control biopsy than diagnosis without an MRI [7]. Higher grading of tumor-suspicious MRI lesions was also associated with disease progression during AS [8]. Moreover, PSA derivatives like PSAD or PSA velocity (PSA increase over a specific time interval; PSAV) can be clinical indicators for tumor reclassification during AS [4]. In the PASS (Canary Prostate Active Surveillance) protocol, PSA kinetics outperformed the baseline PSA level for the prediction of ISUP reclassification during AS [9].

However, optimal schedules and indicators for control biopsies on AS have not yet been established, and monitoring via repeated biopsies has limited acceptance among patients. Furthermore, the informative value of tissue biopsies might be hindered by the heterogeneity and multifocality of PCa. In contrast, liquid biopsies, such as urine and blood, can be easily repeated over time and provide a more global assessment of PCa than tissue biopsies [10,11]. Particularly, urine is an attractive source for biomarker research as it can be obtained non-invasively. Due to the anatomic proximity of the urinary system to the prostate, PCa-derived cells and extracellular vesicles (EV) are released into the urine [10,11,12]. EVs are small membranous structures (~30–200 nm in diameter) secreted by most cells into their surrounding microenvironment with a particular abundance in body fluids [13,14]. EVs transport various cargo (e.g., DNA, RNA, proteins, lipids) for intercellular communication and thus reflect the molecular profile of their parent cells [14,15]. For instance, EVs can mediate the therapeutic resistance of cancer cells [15,16]. The EV cargo is protected from degradation by the surrounding lipid bilayer, which makes EV a useful source for biomarker research from biofluids, particularly for transcript markers [14].

To date, there are no established molecular biomarkers for AS monitoring and the prediction of risk reclassification at the control biopsy [12,17]. Studies evaluating the use of transcript markers from urine for AS monitoring are scarce, as most studies deal with optimizing the selection criteria for AS enrollment. Tao et al. and Connell et al. both constructed biomarker models based on transcripts from urinary EV (uEV) that could prognosticate the time to reclassification for men initiated on AS [18,19]. Using whole urine samples, the commercially available PCA3 test (Progensa PCA3), which is only approved for biopsy decision-making in men with unknown PCa and a previous negative biopsy, has been frequently evaluated for AS monitoring [20,21,22,23,24,25]. PCA3 is a long non-coding RNA (lncRNA), which is upregulated in PCa and associated with PCa aggressiveness [26]. However, in the setting of AS monitoring, urinary PCA3 has demonstrated varying degrees of predictive power for risk reclassification [20,21,22,23,24,25]. Still, molecular biomarkers possess the potential to support and improve the findings from clinical and imaging parameters [27].

Therefore, we investigated the predictive potential of selected transcripts from uEV from 72 PCa patients on AS for PCa risk reclassification at the control biopsy. Based on our own previous data [28,29,30,31,32] and extensive literature search (selected reviews [33,34]), the evaluated transcripts included 29 mRNAs and lncRNAs. The selected transcripts are known to be differentially expressed in PCa (e.g., AMACR, ERG, PCA3, PSMA) and/or are functionally involved in important biological processes, such as apoptosis (e.g., BCL2, BCL-XL, MCL1), cell cycle (e.g., CCND1), and transcription (e.g., STAT3, STAT5A, STAT5B) [28,29,30,31,32,33,34]. Via a combination of uEV transcript markers with clinical and imaging parameters, we aimed to develop a multi-marker panel to further improve the prediction of PCa risk reclassification.

## 2. Materials and Methods

### 2.1. Patient Cohort

Patients with known low- or early intermediate-risk PCa on AS presenting for MRI and subsequent control biopsy were prospectively included between April 2018 and February 2023 in this study. Patients with prior treatment of PCa (radiotherapy, focal therapy, androgen deprivation therapy) or on α-reductase inhibitors were excluded. The study was approved by the institutional review board of the Technische Universität Dresden (EK346082016 & EK123032019) and conducted according to the Declaration of Helsinki. Written informed consent was obtained from all patients before study inclusion. Study data were collected and managed using REDCap electronic data-capture tools (version 14.0.31) hosted at our institutions. REDCap (Research Electronic Data Capture) is a secure, web-based software platform designed to support data capture for research studies [35,36].

The initial PCa diagnosis was confirmed histopathologically by prostate biopsy or detected as incidental PCa during the transurethral resection of the prostate tissue (TUR-P). MRI was not mandatory for the initial diagnosis. Overall, 72 patients scheduled for control biopsy on AS were included in the present analysis.

PSAD (ratio of serum PSA level and prostate volume based on MRI or transabdominal ultrasound examination) and PSAV (change of initial PSA [iPSA] at first diagnosis to PSA at control biopsy in relation to the interval between both time points) were determined. Bi- or multi-parametric MRIs (mpMRI) were performed in-house or in ambulant radiology offices before the control biopsy. MRIs were evaluated according to the Prostate Imaging Reporting and Data System (PI-RADS) v2.1, and the maximal PI-RADS value (maxPI-RADS) was determined. All patients underwent a 12-core systematic biopsy according to an in-house scheme. In patients with maxPI-RADS ≥ 3 lesions in MRI, an additional targeted MRI/ultrasound fusion biopsy was performed. Two to four cores were taken per lesion depending on the lesion size.

Risk reclassification of PCa was defined as an upgrading of ISUP to ISUP ≥ 2 and/or increase of PSA > 10 ng/mL in case of constant ISUP 1. All other patients (tumor-free biopsy regardless of PSA value and detection of ISUP 1 with PSA ≤ 10 ng/mL) were considered stable.

### 2.2. Collection and Processing of Urine Samples

Following digital-rectal examination (DRE) and at a median of 13 days (range from 2 to 29 days) before biopsy, urine specimens were collected as described previously [37]. Urine samples were immediately centrifuged for 5 min at 870× *g* and 4 °C, and 8 mL of cell-free supernatant were stored at −80 °C until uEV isolation.

After thawing and equilibration at 37 °C for 10 min in order to increase uEV recovery [38], cell-free urine supernatants were centrifuged for 5 min at 3200× *g* and room temperature. Next, the miRCURY Exosome Cell/Urine/CSF Kit (Qiagen, Hilden, Germany) was used to isolate uEV. Briefly, 2.1 mL of EV precipitation buffer were added to 7 mL urine supernatant followed by incubation overnight at 4 °C and further processing according to the manufacturer’s protocol. Ultimately, precipitated uEV were lysed in 700 µL QIAzol Lysis Reagent (Qiagen) and stored at −80 °C until RNA isolation.

For subsequent uEV characterization, two additional urine samples were processed per isolation round. Instead of QIAzol Lysis Reagent, precipitated uEV were resuspended in 70 µL XE buffer (Qiagen), of which 53.4 µL were lysed with 6.6 µL RIPA buffer 10× (Cell Signaling, Danvers, MA, USA) containing a Protease Inhibitor Cocktail 100× (Sigma–Aldrich, Taufkirchen, Germany), followed by shaking for 30 min at 4 °C. The uEV protein lysate and the remaining intact uEV in XE buffer were stored at −80 °C until further use.

### 2.3. Characterization of uEV Control Samples

In order to establish the presence of small uEV, selected control samples were further characterized as recommended by the 2023 guidelines on “Minimal Information for Studies of Extracellular Vesicles” (MISEV2023) by the International Society for Extracellular Vesicles [13].

Nanoparticle tracking analysis was performed using the ZetaView^®^ system (Particle Metrix, Inning am Ammersee, Germany) according to the manufacturer’s instructions to determine the size distribution of the isolated uEV in XE buffer.

Furthermore, the presence of EV markers (CD9, CD63, TSG101) and the absence of a cellular marker (calnexin) were determined by Western Blot analysis. Firstly, the protein concentration of uEV lysates was quantified with a BCA assay (Thermo Fisher Scientific, Darmstadt, Germany) on a Berthold Mithras LB940 microplate reader (Berthold Technologies, Bad Wildheim, Germany) according to the manufacturer’s instructions. Next, lysates were mixed with NuPAGE LDS sample buffer (Thermo Fisher Scientific) either with 5% β-mercaptoethanol (reducing conditions for calnexin, CD9, TSG101) or without it (non-reducing conditions for CD63) followed by incubation at 70 °C for 10 min. Lysates (median protein amount 21 µg) were then separated on 4–12% NuPAGE Bis-Tris protein gels using the NuPAGE MES SDS running buffer as well as Spectra Multicolor Broad Range Protein Ladder and MagicMark XP Western Protein Standard for protein molecular weight estimation (all from Thermo Fisher Scientific). Lysates of the PCa cell lines PC-3 and LNCaP (CRL-1435 & CRL-1740, ATCC, Manassas, VA, USA) were used as cellular control (for details about cultivation and harvest of cell lines see [39,40]). Subsequently, proteins were transferred onto nitrocellulose membranes using the iBlot gel transfer device (all from Thermo Fisher Scientific). After blocking with non-fat dry milk for 2 h, membranes were probed with primary antibodies against calnexin (1:1000; clone C5C9, Cell Signaling), CD9 (1:500; clone EPR2949, Abcam, Cambridge, UK), CD63 (1:5000; clone TS63, Thermo Fisher Scientific), and TSG101 (1:1000; clone 4A10, BIOZOL Diagnostica, Eching, Germany) overnight at 4 °C. Following several washing steps, the membranes were incubated with the appropriate secondary HRP-linked antibody (polyclonal rabbit anti-mouse immunoglobulin: 1:10,000, P026002-2; polyclonal swine anti-rabbit immunoglobulin: 1:5000-1:10,000, P021702-2; both Agilent Technologies, Waldbronn, Germany) for 1.5 h at room temperature. Finally, immunoreactive bands were visualized using the WesternBright Sirius Chemiluminescent Detection Kit (Advansta, San Jose, CA, USA) and captured with the MicroChemi 4.2 system (DNR Bio-Imaging Systems, Jerusalem, Israel).

### 2.4. RNA Isolation, cDNA Synthesis, Pre-Amplification, and Quantitative Polymerase Chain Reaction (qPCR)

Total RNA from uEV in QIAzol Lysis Reagent was extracted using the miRNeasy Micro Kit (Qiagen) according to the manufacturers’ recommendations. RNA quantity and quality were evaluated using the NanoDrop 2000c spectrophotometer (PEQLAB, Erlangen, Germany).

Synthesis of cDNA was conducted with up to 500 ng of total RNA using SuperScript III Reverse Transcriptase, RNaseOUT ribonuclease inhibitor (both Thermo Fisher Scientific), and random hexamer primers (Sigma–Aldrich). Afterwards, a pre-amplification step was performed by means of TaqMan Gene Expression Assays (Appendix A) and the TaqMan PreAmp Mastermix (both Thermo Fisher Scientific) according to the manufacturer’s recommendations. For quantification of transcripts by qPCR, each qPCR reaction contained 1 µL of pre-amplified cDNA (1:20) or nuclease-free water (negative control), 0.5 µL of specific TaqMan Gene Expression Assay (Appendix A), 5 µL of GoTaq Probe qPCR Master Mix (Promega, Mannheim, Germany), and 3.5 µL of nuclease-free water. Two independent qPCR measurements were run per sample using the LightCycler 480 Real-Time PCR System (Roche Diagnostics, Mannheim, Germany) with the following conditions: initial denaturation at 95 °C for 10 min followed by 45 cycles at 95 °C for 15 s and 60 °C for 60 s. Measurements were repeated if the averaged crossing point (CP) values deviated > 0.5. CP values ≥ 35 (detection limit) indicated negative samples. Averaged CP values were then used to calculate relative transcript expression by the ΔΔCP method with normalization to the geometric CP mean of the reference RNAs PPIA, RPLP0, and TBP (geoM).

### 2.5. Statistical Analysis

Statistical analyses were conducted using GraphPad Prism 10.2.2 (GraphPad Software, San Diego, CA, USA) and IBM SPSS Statistics 29.0.0.0 (IBM, Armonk, NY, USA). The Mann–Whitney U test (continuous variables) or Fisher’s Exact test (categorized variables) was used for two group comparisons. The Kruskal–Wallis test (continuous variables) or Chi-square test (categorized variables) was used to evaluate associations among more than two groups. In order to assess the predictive potential of clinical and transcript markers alone or in combination, the area under the curve (AUC) values, including their 95% confidence intervals (95% CI), were calculated by receiver operating characteristic (ROC) curve analysis. Following calculation of the Youden index to identify the optimal cutoff value for risk reclassification, the sensitivity (SNS), specificity (SPC), positive predictive value (PPV), negative predictive value (NPV), positive likelihood ratio (pLR), negative likelihood ratio (nLR), and accuracy (ACC) were determined for selected markers alone or in combination. Univariate logistic regression was performed to determine the odd ratios (OR) of clinical and transcript markers that had shown a predictive potential in ROC curve analysis (*p* < 0.1). A multivariate logistic regression (forward stepwise) was then implemented to identify independent predictors, which were subsequently combined. To test the predictive value of marker combinations, sum scores were generated according to whether each of the included markers predicted risk reclassification (1) for a patient or not (0). These sum scores were then evaluated via ROC curve analysis as described above. A *p* value < 0.05 was considered statistically significant, whereas *p* values ≥ 0.05 and <0.1 indicated a statistical trend.

## 3. Results

### 3.1. Characteristics of Patient Cohort

Overall, 72 patients on AS were included in this study, of which 31 (43%) later showed a risk reclassification at the control biopsy. The median iPSA at the first diagnosis of PCa and subsequent AS initiation was similar in patients with later stable and reclassified disease (Table 1). The time from the first diagnosis (2006–2022) to the control biopsy (2018–2023) did not differ significantly between both groups (stable: 1.4 years [range 0.3–10.3 years] versus reclassified: 1.6 years [range 0.5–12.2 years]). Overall, 47 (65%) participants were within 2 years of their diagnosis. At the time of the control biopsy, patients with tumor reclassification were significantly older and showed higher median PSA, PSAD, and PSAV values. Moreover, reclassified patients exhibited more often a PSA increase >10 ng/mL, a PSAD > 0.2 ng/mL^2^, and a positive PSAV. Furthermore, patients with a reclassified PCa presented more often tumor-suspicious lesions in MRIs (maxPI-RADS ≥ 4) at the control biopsy than patients with stable disease (65% versus 29%; *p* = 0.004). None of the patients had lymph node or distant metastases at the time point of risk reclassification. Eventually, 25 patients with PCa risk reclassification underwent definitive treatment (radical prostatectomy *n* = 16, radiation therapy *n* = 8, focal therapy *n* = 1). Four patients without risk reclassification (ISUP 1) opted for curative treatment and underwent radical prostatectomy. Six patients remained on AS despite risk reclassification. Of the 20 patients undergoing radical prostatectomy, four (20%) presented locally advanced PCa (pT3). Only three of the stable patients upgraded from ISUP 1 to ISUP 2, whereas the others showed no further ISUP upgrading in the prostatectomy specimen.

### 3.2. Characterization of Isolated uEV

Before the control biopsy, urine samples were prospectively collected from the included patients and utilized for the uEV isolation. Overall, 10 control samples were used for further uEV characterization. The median size of the isolated uEV was 132 ± 17 nm. Furthermore, the presence of the EV markers CD9, CD63, and TSG101 could be confirmed (Figure 1). In contrast, the cellular marker calnexin was absent in all uEV samples but highly positive in PC-3 and LNCaP PCa cells (Figure 1). Overall, the isolated uEV could be identified as a small EV < 200 nm without cellular contamination.

### 3.3. Deregulation of uEV Transcripts in Patients with Risk Reclassification

Overall, 29 selected transcripts were analyzed by qPCR in uEV. Of all investigated transcripts, only AMACR was significantly upregulated by 1.4-fold in patients with risk reclassification (*p* = 0.024; Appendix A, Figure 2A). Furthermore, HPN, PCA3, and PCAT29 were, per trend, increased in uEV from patients with risk reclassification by up to 1.7-fold (*p* < 0.1; Appendix A, Figure 2). In contrast, MALAT1 was, per trend, downregulated by about 1.5-fold (*p* = 0.099; Appendix A, Figure 2C). All other transcripts showed no differential expression in uEV from patients with risk reclassification compared to patients with stable disease (Appendix A).

### 3.4. Association of Clinical and Transcript Markes with Biopsy Tissue Grading

Of the clinical parameters, only PSAV and maxPI-RADS were significantly associated with biopsy tissue grading (Table 2). Patients with ISUP grading groups 2–5 showed the highest PSAV values and the highest percentage of MRI lesions with maxPI-RADS 4 + 5. Although PSA and PSAD values also increased in the higher ISUP grading groups, this association was not significant. Furthermore, the transcripts AMACR, HPN, and PCAT29 were, per trend, elevated in patients with advanced ISUP grading (Table 2).

### 3.5. Predictive Potential of Clinical and Transcript Markers

Next, the predictive potential of clinical parameters and uEV transcripts regarding the detection of risk reclassification was investigated. The clinical parameters PSA, PSAD, PSAV, and maxPI-RADS showed moderate AUC values (0.681–0.747, *p* < 0.05) for indicating PCa risk reclassification (Figure 3 and Table 3). ACC values for PSA (cutoff >10 ng/mL), PSAD (cutoff > 0.2 ng/mL^2^), PSAV (cutoff > 0 ng/mL/year), and maxPI-RADS (cutoff ≥ 4) ranged from 64% to 68% (Table 3). For PSA and PSAD, cutoff values according to the PRIAS project were used [5].

Furthermore, ROC curve analyses revealed a potential for the prediction of PCa risk reclassification for the uEV transcripts AMACR (AUC = 0.655, *p* = 0.025), HPN (AUC = 0.626, *p* = 0.068), MALAT1 (AUC = 0.614, *p* = 0.098), PCA3 (AUC = 0.617, *p* = 0.091), and PCAT29 (AUC = 0.627, *p* = 0.067) (Figure 3 and Table 3). By using the optimal cutoff values based on the Youden index, the ACC values of these transcripts ranged from 61% to 68% (Table 3) and were, thus, similar to those of the clinical parameters. All other transcripts were without predictive potential (Appendix A).

### 3.6. Increased Predictive Potential of Combined Clinical and Transcript Markers

Using the above-reported cutoff values, the clinical and transcript markers with before-proven predictive potential (PSA, PSAD, PSAV, maxPI-RADS, AMACR, HPN, MALAT1, PCA3, and PCAT29) were analyzed by univariate logistic regression. All markers were significantly associated with risk reclassification in univariate analysis (Table 4).

Next, a multivariate logistic regression was conducted in a forward stepwise fashion in order to identify independent predictors, whose combination could possess an enhanced predictive power. Ultimately, PSAD, maxPI-RADS, AMACR, MALAT1, and PCAT29 were identified as independent predictors (Table 4).

Subsequently, the predictive potential of the combined independent clinical and transcript markers (2C-3T-Score: PSAD, maxPI-RADS, AMACR, MALAT1, and PCAT29) was investigated and compared to combinations of both clinical parameters (2C-Score: PSAD and maxPI-RADS) and all three transcript markers (3T-Score: AMACR, MALAT1, and PCAT29). The combination of all five markers (2C-3T-Score) resulted in a highly improved AUC value of 0.867 (*p* < 0.001) and ACC of 85% compared to the individual markers, as well as to the 2C- and 3T-Score (Figure 4 and Table 5). The improved discriminatory power was mostly accompanied by higher SNS, SPC, PPV, NPV, and pLR, as well as by lower nLR values (Table 5).

Furthermore, the ORs for the various combinations were calculated by univariate logistic regression (Table 6). The highest OR was observed for the 2C-3T-Score when patients presented at least three of the five risk factors included in this model.

Using the 2C- and 3T-Score alone, 39% and 38% of control biopsies could have been omitted in case of a negative score, while 11% and 4% of reclassified PCa would have been missed, respectively (Table 6). However, in the case of a negative 2C-3T-Score, the control biopsy could have been dispensed in 47% of the cases, while the missing rate of reclassified PCa would only have been 6%. Furthermore, 18% and 19% of the patients with a positive 2C- and 3T-Score but stable disease would have been recommended for a control biopsy, respectively. In the case of a positive 2C-3T-Score, that would only have applied to 10% of the patients with stable disease.

## 4. Discussion

AS protocols for patients with low- and early intermediate-risk PCa incorporate serial PSA measurements, imaging, and prostate biopsy to identify disease progression [4]. However, there are currently no valid clinical parameters or biomarkers that can indicate PCa risk reclassification upfront in order to reduce morbidity associated with biopsy [17]. Molecular biomarkers could support and improve the findings from clinical and imaging parameters [27]. Therefore, we evaluated selected uEV transcript markers, as well as clinical and imaging parameters, to predict PCa risk reclassification in a prospective cohort of 72 patients undergoing MRI-guided control biopsy during AS.

### 4.1. Predictive Potential of the 2C-3T-Score

Statistical analyses revealed a predictive potential for the uEV transcripts AMACR, HPN, MALAT1, PCA3, and PCAT29 (AUC = 0.614–0.655, ACC = 61–68%, OR = 3.4–5.0). The clinical parameters PSA, PSAD, PSAV, and maxPI-RADS showed AUC, ACC, and OR values for indicating PCa risk reclassification of 0.681–0.747, 64–68%, and 4.4–7.0, respectively. Via multivariate regression analysis, a model including AMACR, MALAT1, PCAT29, PSAD, and maxPI-RADS was identified with the best predictive potential. Combining these markers into the 2C-3T-Score resulted in a highly improved AUC of 0.867, an ACC of 85%, and an OR of 32.8, thus surpassing the predictive power of the individual markers.

A survey distributed to clinicians and patients demonstrated that the majority (91–93%) was comfortable with a non-invasive biomarker test that could replace the AS biopsy, yet only 65%–78% would accept false negative rates of 5–20% [41]. Our 2C-3T-Score distinguished well between reclassified and stable diseases with a high sensitivity of 87% and an NPV of 90%. In the case of a negative 2C-3T-Score, 47% of control biopsies could have been omitted, while only 6% of reclassified PCa would have been missed. Therefore, patients might benefit from a multimodal strategy integrating clinical features and biomarker profiles to predict the reclassification risk. Urinary biomarkers can provide a more global assessment of the tumor, whereas mpMRI also offers spatial information for possible biopsy targets, which can subsequently be used for a targeted MRI/ultrasound fusion biopsy in case of a positive 2C-3T-Score. However, in cases where mpMRI is not readily available, even the molecular 3T-Score might be useful for the prediction of risk reclassification since its predictive power was higher than for the clinical 2C-Score (AUC: 0.811 versus 0.735, ACC: 76% versus 71%, OR: 18.0 versus 6.2).

### 4.2. Predictive Potential of Clinical and Imaging Parameters

To date, clinico-pathological parameters have mostly been evaluated regarding their predictive potential for risk reclassification of PCa patients on AS. In our cohort, we could demonstrate that PSAD (>0.2 ng/mL^2^) and maxPI-RADS (≥4) were independent clinical predictors for PCa risk reclassification in control biopsy. These findings are consistent with the results of the European PRIAS study. Herein, Luiting et al. identified MRI findings (PI-RADS 3: hazard ratio [HR] 2.46, PI-RADS 4: HR 3.39, PI-RADS 5: HR 4.95), PSAD (HR 1.2), and percentage-positive cores on the last biopsy (HR 1.16) as independent predictors of PCa risk reclassification [42]. Higher PSAD levels reliably indicated risk reclassification at a confirmatory biopsy among men with low-grade PCa and negative MRI [43]. Greenberg et al. investigated 70 patients with AS who obtained ≥2 mpMRI. Men experiencing an upgrade of MRI also had an increased risk of PCa reclassification and demonstrated a shorter reclassification-free survival than patients without MRI progression [44]. In our cohort, we evaluated the control imaging nearest to the AS control biopsy. Since patients were included regardless of the initial MRI at first diagnosis, we could not evaluate a progression in the MRI findings, such as the change in the number or size of tumor-suspicious lesions.

We could show that the combination of MRI and PSAD in the 2C-Score improved the prediction of risk reclassification with an AUC of 0.735, and the presence of at least one risk factor increased the probability of reclassification by 6.2-fold. In the case of a negative 2C-Score, 39% of prostate biopsies could have been omitted while missing 11% of reclassified PCa. Felker et al. also reported improved diagnostic accuracy, when mpMRI was combined with a PSAD of >0.15 ng/mL^2^ (AUC 0.87 vs. 0.63) [45]. Luzzago et al. retrospectively observed that patients on AS with a stable MRI and PSAD < 0.15 ng/mL^2^ could safely be monitored by an annual MRI alone and omit invasive biopsies [46].

### 4.3. Predictive Potential of Urinary Transcript Markers

In contrast to the clinico-pathological parameters of the 2C-3T-Score, the uEV transcript levels of *AMACR* (alpha-methylacyl-CoA racemase, also known as P504S), MALAT1 (metastasis-associated lung adenocarcinoma transcript 1), and PCAT29 (prostate cancer-associated transcript 29) have not been evaluated in the setting of AS monitoring so far. All three markers are known to be deregulated in cancer, including PCa [28,47,48,49]. Functionally, AMACR is involved in the metabolism of fatty acids, and the immunohistochemical staining of the AMACR protein is routinely used by pathologists on biopsy specimens to achieve definitive PCa diagnosis [47]. The lncRNAs MALAT1 and PCAT29 act as transcriptional regulators for numerous genes and, thus, influence cellular functions, such as proliferation, migration, and invasion [49,50,51]. Although mRNAs and lncRNAs differ in their biological functions (encoding the amino acid sequence of proteins versus regulating other transcripts at the transcriptional, post-transcriptional, or epigenetic level) [52], both can serve as diagnostic and/or predictive biomarkers when differentially expressed in cancer [33,34].

Studies evaluating other transcript markers from uEV for AS monitoring are also limited. Tao et al. identified a diagnostic model combining three lncRNAs from uEV for detecting high-grade PCa with ISUP ≥ 2 [19]. In a prospective AS cohort (*n* = 182), this marker combination was an independent prognosticator of time to reclassification (HR 2.1). Connell et al. constructed a risk classifier based on the expression levels of 36 uEV transcripts (i.a., AMACR, HPN, PCA3) that were associated with time to reclassification (HR 8.23) for men on AS (*n* = 87) [18]. In contrast to our work, these two prognostic studies analyzed the uEV biomarkers at the time of AS enrollment due to PCa diagnosis. The commercially available ExoDx Prostate Intelliscore test, which determines the expression levels of PCA3 and ERG in uEV, has, to date, only been investigated to stratify PCa patients for AS enrollment [53] but not for AS monitoring.

By using whole urine instead of uEV, the commercially available PCA3 test has frequently been evaluated for AS monitoring. In a study by Tosoian et al. (*n* = 294), urinary PCA3 levels were only slightly predictive of risk reclassification during AS (AUC = 0.589) [24] but added predictive value to a multivariate model, including age, PSAD, and risk strata (AUC 0.740 vs. 0.700) in a follow-up study (*n* = 260) [25]. Interestingly, this latter association was observed regardless of whether urinary PCA3 was determined early (1–3 years) or late (4–6 years) in the course of AS [25]. Fradet et al. observed that an elevated PCA3 score from urine collected at the first control biopsy after AS initiation (*n* = 90) could predict PCa risk reclassification in subsequent control biopsies during a median follow-up of 7 years [20].

Using the PASS cohort (*n* = 387), Lin et al. showed that the urinary scores of PCA3 and the fusion transcript TMPRSS2:ERG were able to predict high-grade disease with AUC values ranging from 0.608 for individual markers to 0.702 for the combination of both markers with serum PSA [21]. However, only 22% of the urine samples were collected at the time of control biopsy, whereas the majority was obtained up to 46 months after biopsy (91% within 1 year). In our study, the control biopsy occurred, at most, 29 days after urine collection, and, thus, we could assess the immediate association between biomarker results and biopsy outcome. In a follow-up study by Newcomb et al., the addition of PCA3 to clinical parameters (serum PSA, core PCa ratio, and prostate size) only provided a slight improvement of AUC (0.753 versus 0.743) for predicting risk reclassification at the first surveillance biopsy (*n* = 552) [22]. In a study (*n* = 154) including men on AS (24%) and those with prior negative biopsy but ongoing PCa suspicion (76%), none of the patients with a normal PCA3 score and negative mpMRI (maxPI-RADS ≤ 3) were diagnosed with a clinically significant PCa on repeat biopsy (NPV = 100%) [23].

In our study, PCA3 was only slightly upregulated in patients with PCa risk reclassification, which in turn resulted a rather weak predictive potential (AUC = 0.617, *p* = 0.091). Although our results were mostly similar to the AUC values for PCA3 alone in the aforementioned studies, one has to keep in mind that these studies used the commercially available PCA3 test for whole urine with normalization to PSA mRNA. In contrast, we quantified PCA3 in uEV with a non-diagnostic TaqMan qPCR assay with normalization to the geoM of PPIA, RPLP0, and TBP mRNA. Furthermore, other methodical criteria, such as cohort composition, the definition of risk reclassification, and the time interval between urine collection and control biopsy, are highly varied, which makes it difficult to compare our results with the discussed studies. Nevertheless, combinations of molecular biomarkers with clinical and imaging features often resulted in an improved prediction of risk reclassification, which we could also demonstrate here.

Accordingly, molecular uEV markers in combination with clinical and imaging parameters could help to optimize monitoring protocols for AS and consequently minimize the number of control biopsies and healthcare costs. Urinary liquid biopsies are obtained non-invasively, can sample biomolecules of the entire prostate, and help to address the multifocality and heterogeneity of PCa. Hence, they could counteract tissue biopsy sampling biases and are theoretically more suitable than tissue-based biopsies for therapy monitoring [10,11]. A disadvantage of liquid biopsies is that potential markers may be influenced by other non-organ-specific components and molecules [11]. For example, other pre-existing diseases, such as urinary tract infections or even tumors, can influence the composition of the urine. In particular, urine-based biomarkers can be further influenced by the size and manipulations (e.g., DRE) of the prostate [12]. In accordance, previous studies have shown that DRE before urine collection resulted in higher biomarker levels (e.g., PCA3) in uEV and, thus, enhanced the analytical performance of the biomarker analysis [54,55].

### 4.4. Strengths and Limitations of the Study

The strength of our study is its prospective study design, which reduces typical confounders of retrospective studies. All patients received an MRI before the control biopsy, and urine was collected and processed according to a standardized protocol shortly before the control biopsy. Consequently, there was a relatively short time interval between sample collection and biopsy (maximum 29 days), so that the detection of risk reclassification was in close temporal relation to the sample collection and assessment compared to other study cohorts. After the control biopsy, patients received a stringent follow-up of at least 1 year.

However, our study also has some limitations. First, we included patients at any time point during AS due to the absence of a homogeneous AS protocol. Patients were included regardless of the number of previous control biopsies and the interval between AS initiation and control biopsy, which could be up to 12 years. However, the majority of the participants (65%) were within 2 years of their AS after PCa diagnosis. Furthermore, the 2C-3T-Score was highly predictive for risk reclassification, whether it was determined early or late in the course of AS. Second, the inclusion criteria for AS enrollment were inhomogeneous, e.g., MRI at AS enrollment was not mandatory, and, thus, longitudinal MRI findings were not available. Therefore, this study might have a higher risk of selection bias. Third, since patients with in-house MRI and MRI performed in ambulatory radiology offices were included, there might be the risk of inter-observer variability regarding the description and reporting of MRI results. We did not conduct a second central review in the case of external MRIs within this study. Fourth, before urine collection, all patients were examined by DRE, which was performed by several physicians with different levels of training. Therefore, the effect of DRE on the yield of prostate-derived molecules and/or uEV in the urine might be inhomogeneous. Fifth, we could not report on baseline or serial liquid biopsy samples. Therefore, our marker model represents only risk reclassification at a specific time point during AS, and we cannot draw conclusions on the prognosis of our patients since long-term follow-up is missing. Lastly, biomarker analysis was only conducted in a test cohort including 72 patients. Further validation of our marker model has to be performed on another independent prospective cohort with more patients.

## 5. Conclusions

To the best of our knowledge, this is the first study to use the expression levels of uEV transcripts to predict PCa risk reclassification at the time of AS control biopsy. Combining established clinical markers (PSAD and maxPI-RADS) with uEV transcript markers (AMACR, MALAT1, and PCAT29) resulted in a highly improved predictive power for indicating PCa risk reclassification. Although based on a small patient cohort, our findings highlight the potential of uEV transcripts in combination with clinical parameters as monitoring markers during AS of PCa patients. Overall, a multimodal approach integrating clinical features and biomarker profiles will help to develop more individualized AS strategies and omit unnecessary invasive diagnostics. However, further validation in a larger patient cohort should be conducted in the future.

## Figures and Tables

**Figure 1 cancers-16-02453-f001:**
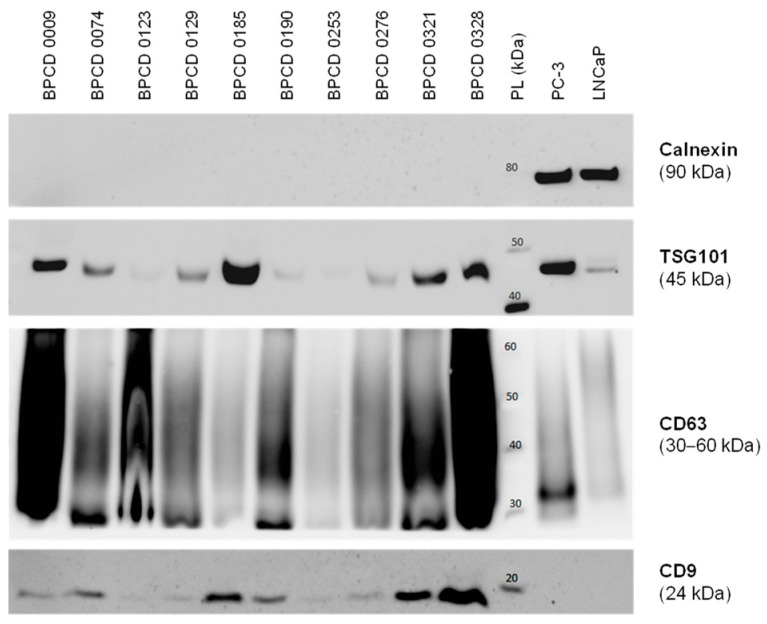
Characterization of isolated uEV by Western blotting. Exemplary Western blots probing for the established EV markers TSG101, CD63, and CD9, as well as the cellular marker calnexin in selected control samples. The PCa cell lines PC-3 and LNCaP served as positive controls for cellular components. PL: protein ladder. The uncropped Western blots are shown in Appendix A.

**Figure 2 cancers-16-02453-f002:**
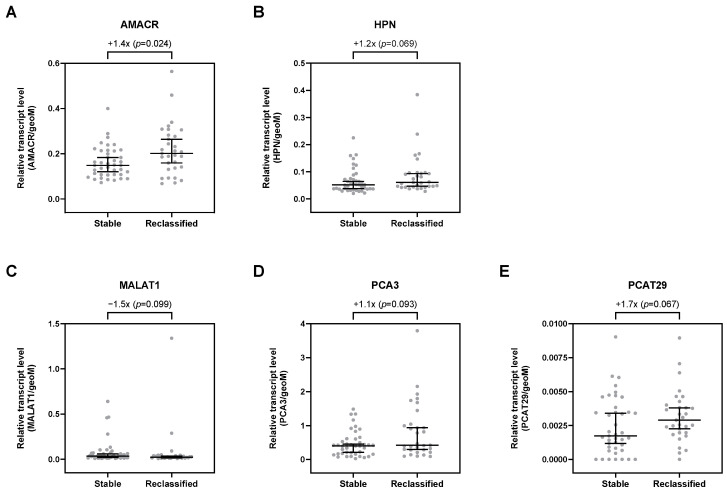
Relative expression levels of (**A**) AMACR, (**B**) HPN, (**C**) MALAT1, (**D**) PCA3, and (**E**) PCAT29 in uEV from stable and reclassified patients at control biopsy. Depicted are the median relative transcript levels ± 95% CI of the evaluated transcripts (normalized to geoM of PPIA, RPLP0, and TBP). *p* values were calculated by the Mann–Whitney U test.

**Figure 3 cancers-16-02453-f003:**
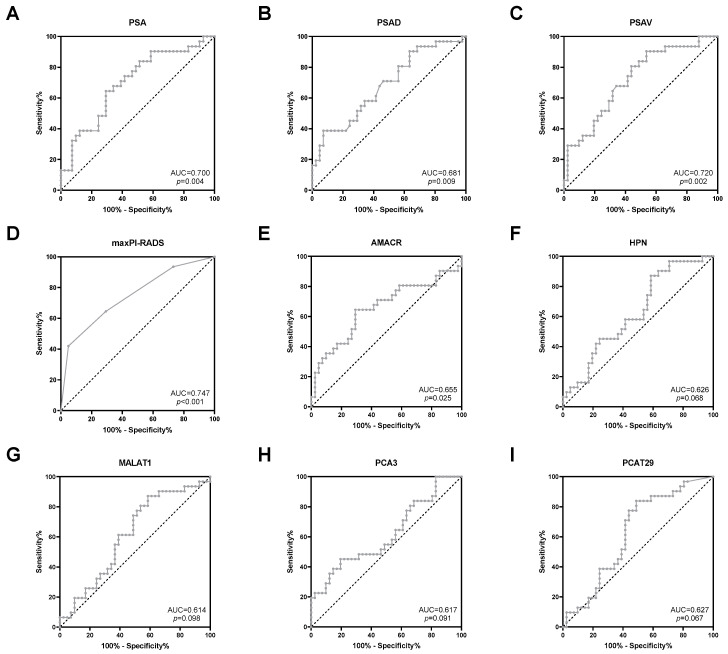
ROC curve analysis of (**A**) PSA, (**B**) PSAD, (**C**) PSAV, (**D**) maxPI-RADS, (**E**) AMACR, (**F**) HPN, (**G**) MALAT1, (**H**) PCA3, and (**I**) PCAT29 to discriminate between stable disease and PCa risk reclassification.

**Figure 4 cancers-16-02453-f004:**
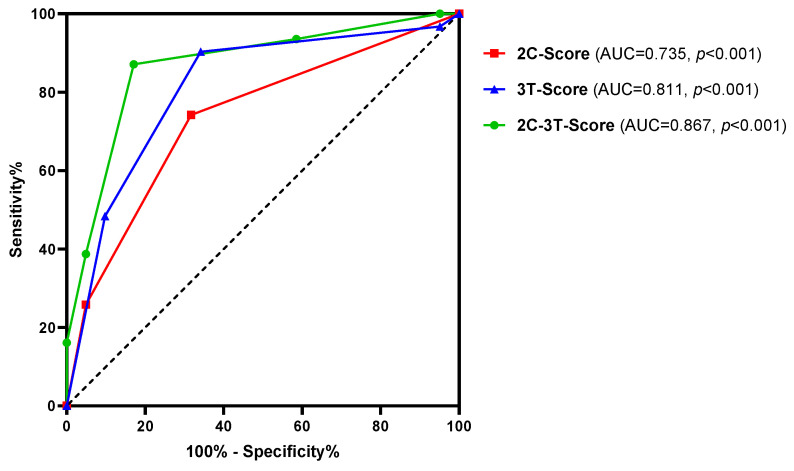
ROC curve analysis of the 2C-3T-Score (PSAD, maxPI-RADS, AMACR, MALAT1, and PCAT29), 2C-Score (PSAD and maxPI-RADS), and 3T-Score (AMACR, MALAT1, and PCAT29) to discriminate between stable disease and PCa risk reclassification.

**Table 1 cancers-16-02453-t001:** Demographic and clinico-pathological characteristics of the patient cohort.

Parameter	Stable *n* = 41	Reclassified *n* = 31	*p* Value
**(1) Status at 1st Diagnosis of PCa & Initiation of AS**			
**Diagnosis of PCa**			NA
Biopsy	26	28	
TUR-P	15	3	
**Median iPSA (ng/mL)**	5.38	5.96	0.207
(Range)	(0.92 to 13.2)	(1.20 to 11.9)	
**ISUP Grading Group**			NA
1 (GS ≤ 6)	37	30	
2 (GS = 3 + 4)	4	1	
**(2) Status at Control Biopsy on AS**			
**Median Time from PCa diagnosis (years)**	1.4	1.6	**0.099**
(Range)	(0.3 to 10.3)	(0.5 to 12.2)	
**Time from PCa diagnosis**			0.321
≤2 years	29	18	
>2 years	12	13	
**Median Age (Years)**	64	70	**0.004**
(Range)	(55 to 81)	(53 to 84)	
**Median PSA (ng/mL)**	4.82	7.54	**0.003**
(Range)	(0.36 to 13.5)	(0.49 to 19.6)	
**PSA**			**0.011**
≤10 ng/mL	38	21	
>10 ng/mL	3 ^a^	10	
**Median PSAD (ng/mL^2^)**	0.108	0.143	**0.008**
(Range)	(0.002 to 0.313)	(0.019 to 0.560)	
**PSAD**			**0.005**
≤0.2 ng/mL^2^	38	20	
>0.2 ng/mL^2^	3	11	
**Median PSAV (ng/mL/year)**	−0.15	0.60	**0.001**
(Range)	(−12.2 to 3.66)	(−3.52 to 5.41)	
**PSAV**			**0.007**
≤0 ng/mL/year	21	6	
>0 ng/mL/year	20	25	
**DRE (cT)**	1 unknown		0.315
Non-Suspicious (cT1)	36	25	
Suspicious (cT2-4)	4	6	
**Median MRI Lesions**	1	2	**0.004**
(Range)	(0 to 3)	(0 to 3)	
**MRI maxPI-RADS**			**0.004** (2 + 3 vs. 4 + 5)
2	11	2	
3	18	9	
4	10	7	
5	2	13	
**Control Biopsy**			**0.032**
Systematic	11	2	
Systematic & Targeted	30	29	
**Median Biopsy Cores**	15	16	0.130
(Range)	(10 to 18)	(12 to 19)	
**Median Positive Biopsy Cores**	0	3	**<0.001**
(Range)	(0 to 3)	(1 to 15)	
**ISUP Grading Group**			NA
Tumor-free	25		
1 (GS ≤ 6) & PSA ≤ 10 ng/mL	16		
1 (GS ≤ 6) & PSA > 10 ng/mL		4	
2 (GS = 3 + 4)		22	
3 (GS = 4 + 3)		1	
4 (GS = 8)		1	
5 (GS ≥ 9)		3	

*p* values were calculated by the Mann–Whitney U test (continuous variables) or the Fisher’s Exact test (categorized variables) and highlighted in bold to indicate a statistical significance or trend. ^a^ Patients with tumor-free control biopsy. GS: Gleason Score, NA: not applicable.

**Table 2 cancers-16-02453-t002:** Selected clinical parameters and uEV transcripts according to biopsy tissue grading.

Parameter	Tumor-Free *n* = 25	ISUP 1 ^a^ *n* = 20	ISUP 2–5 *n* = 27	*p* Value
Median PSA (ng/mL)	4.54	5.71	7.27	0.137
Median PSAD (ng/mL^2^)	0.107	0.111	0.143	0.113
Median PSAV (ng/mL/year)	−0.18	0.43	0.55	**0.046**
MRI maxPI-RADS				**0.001**
2 + 3	20	12	8	
4 + 5	5	8	19	
AMACR (×10^−3^)	140.00	163.70	208.00	**0.078**
HPN (×10^−3^)	45.22	58.22	60.53	**0.065**
PCAT29 (×10^−3^)	1.72	1.89	3.33	**0.067**

Depicted are the median PSA, PSAD, PSAV, and transcript levels (normalized to geoM of PPIA, RPLP0, and TBP), as well as the patient distribution for maxPI-RADS in each group. *p* values were calculated by the Kruskal–Wallis test (continuous variables) or the Chi-square test (categorized variables) and highlighted in bold to indicate a statistical significance or trend. ^a^ ISUP grading group irrespective of PSA value at control biopsy.

**Table 3 cancers-16-02453-t003:** Predictive power of selected clinical parameters and uEV transcripts to discriminate between stable disease and PCa risk reclassification.

Parameter	PSA	PSAD	PSAV	maxPI-RADS	AMACR	HPN	MALAT1	PCA3	PCAT29
AUC (95% CI)	0.700 (0.578–0.823)	0.681 (0.557–0.806)	0.720 (0.602–0.838)	0.747 (0.631–0.863)	0.655 (0.522–0.789)	0.626 (0.497–0.755)	0.614 (0.483–0.745)	0.617 (0.483–0.750)	0.627 (0.497–0.757)
*p* Value	**0.004**	**0.009**	**0.002**	**<0.001**	**0.025**	**0.068**	**0.098**	**0.091**	**0.067**
Cutoff	>10 ng/mL	>0.2 ng/mL^2^	>0 ng/mL/year	≥4	>185.0 × 10^−3^	>40.8 × 10^−3^	<45.7 × 10^−3^	>718.0 × 10^−3^	>1.7 × 10^−3^
SNS (%)	32.3	35.5	80.6	64.5	64.5	87.1	87.1	45.2	83.9
SPC (%)	92.7	92.7	51.2	70.7	70.7	41.5	41.5	80.5	48.8
PPV (%)	76.9	78.6	55.6	62.5	62.5	52.9	52.9	63.6	55.3
NPV (%)	64.4	65.5	77.8	72.5	72.5	81.0	81.0	66.0	80.0
pLR	4.409	4.849	1.653	2.204	2.204	1.488	1.488	2.315	1.637
nLR	0.731	0.696	0.378	0.502	0.502	0.311	0.311	0.681	0.331
ACC (%)	66.7	68.1	63.9	68.1	68.1	61.1	61.1	65.3	63.9

Established cutoff values were used for PSA and PSAD [5], or optimal cutoff values were determined based on the Youden index. *p* values were calculated by ROC curve analysis and highlighted in bold to indicate a statistical significance or trend.

**Table 4 cancers-16-02453-t004:** Univariate and multivariate logistic regression of selected clinical parameters and uEV transcripts, which had shown a predictive potential in ROC curve analysis (*p* < 0.1), with PCa risk reclassification as the dependent variable.

Parameter	Univariate Analysis			Multivariate Analysis		
	OR	95% CI	*p* Value	OR	95% CI	*p* Value
PSA (reference ≤10 ng/mL)	6.03	1.49 to 24.36	**0.012**			
PSAD (reference ≤0.2 ng/mL^2^)	6.97	1.74 to 27.88	**0.006**	7.17	1.15 to 44.61	**0.035**
PSAV (reference ≤0 ng/mL/year)	4.38	1.48 to 12.90	**0.007**			
maxPI-RADS (reference ≤3)	4.39	1.62 to 11.91	**0.004**	4.09	1.12 to 14.86	**0.033**
AMACR (reference ≤185.0 × 10^−3^)	4.39	1.62 to 11.91	**0.004**	8.00	2.13 to 30.10	**0.002**
HPN (reference ≤40.8 × 10^−3^)	4.78	1.41 to 16.20	**0.012**			
MALAT1 (reference ≥45.7 × 10^−3^)	4.78	1.41 to 16.20	**0.012**	4.52	1.00 to 20.45	**0.050**
PCA3 (reference ≤718.0 × 10^−3^)	3.40	1.19 to 9.68	**0.022**			
PCAT29 (reference ≤1.7 × 10^−3^)	4.95	1.59 to 15.43	**0.006**	4.04	0.96 to 17.06	**0.057**

Multivariate analysis was performed in a forward stepwise fashion using all parameters listed for univariate analysis. *p* values indicating a statistical significance or trend are depicted in bold.

**Table 5 cancers-16-02453-t005:** Predictive power of various combinations of selected clinical parameters and uEV transcripts to discriminate between stable disease and PCa risk reclassification.

Parameter	2C-Score	3T-Score	2C-3T-Score
AUC (95% CI)	0.735 (0.616 to 0.854)	0.811 (0.707 to 0.916)	0.867 (0.779 to 0.956)
*p* Value	**<0.001**	**<0.001**	**<0.001**
Cutoff ^a^	1 of 2	2 of 3	3 of 5
SNS (%)	74.2	90.3	87.1
SPC (%)	68.3	65.9	82.9
PPV (%)	63.9	66.7	79.4
NPV (%)	77.8	90.0	89.5
pLR	2.340	2.645	5.101
nLR	0.378	0.147	0.156
ACC (%)	70.8	76.4	84.7

Optimal cutoff values were determined based on the Youden index. *p* values were calculated by ROC curve analysis and highlighted in bold to indicate statistical significance. ^a^ At least X of Y risk factors must be present to indicate PCa risk reclassification.

**Table 6 cancers-16-02453-t006:** Univariate logistic regression for various combinations of selected clinical parameters and uEV transcripts with PCa risk reclassification as the dependent variable.

Parameter	Stable (*n*)	Reclassified (*n*)	OR	95% CI	*p* Value
**2C-Score**					**<0.001**
Negative (0 risk factors)	28 (39%)	8 (11%)	1		
Positive (≥1 risk factors)	13 (18%)	23 (32%)	6.19	2.19 to 17.51	
**3T-Score**					**<0.001**
Negative (<2 risk factors)	27 (38%)	3 (4%)	1		
Positive (≥2 risk factors)	14 (19%)	28 (39%)	18.00	4.65 to 69.74	
**2C-3T-Score**					**<0.001**
Negative (<3 risk factors)	34 (47%)	4 (6%)	1		
Positive (≥3 risk factors)	7 (10%)	27 (38%)	32.79	8.69 to 123.76	

*p* values indicating a statistical significance are depicted in bold.

## Data Availability

The original contributions presented in the study are included in the article/Appendix A, further inquiries can be directed to the corresponding author.

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
