# Peer review of "Transcript Markers from Urinary Extracellular Vesicles for Predicting Risk Reclassification of Prostate Cancer Patients on Active Surveillance"

_cancers, 2024, doi:10.3390/cancers16132453_

Round 1
Reviewer 1 Report
Comments and Suggestions for Authors
I am sending the authors of the manuscript comments and ask them to take them into account:
1. all abbreviations should be explained when first used (PSA in the abstract),
2. the introduction is too long and needs to be condensed,
3. in what years was the study conducted?
4. what was the period between diagnosis and the start of the study?
5. were the patients previously treated - if so, how?
6. some patients had TUR-P, so when was this procedure performed and when was the urine sample taken - I ask because TUR-P can affect the generation of EVs,
7. I congratulate the authors on the correct characterization of EVs - finally, someone follows MISEV,
8. I have no critical comments about the results - they are extremely carefully prepared,
9. The discussion is also written with great caution, I would be safer in the place of the authors to draw far-reaching conclusions from a study on such a small group of patients,
10. the references need to be expanded:
https://doi.org/10.3390/cells11182913
https://doi.org/10.3390/cancers12113292
https://doi.org/10.3390/cancers13153791
https://doi.org/10.3390/cancers16091717
Reviewer 2 Report
Comments and Suggestions for Authors
This study is very interesting; however, the presentation does not do it justice. The write-up is confusing and chaotic. Here are some suggestions for improvement:
Intro-Clarify why 29 genes/lncRNAs were chosen. What is the justification for this selection? The authors should explain this clearly.
Methods- There is a lack of information regarding the cell lines used as controls. Please include detailed methods for this.
Introduce "CP" before using the abbreviation to ensure clarity.
Results- Explain why only 5 expression levels are shown. What is the justification for this choice?
Discussion- The section currently combines mRNA levels and lncRNAs, which is confusing. Use subsections to separate these topics and discuss the clinical significance of mRNAs and lncRNAs in depth.
Comments on the Quality of English LanguageN/A
Round 2
Reviewer 1 Report
Comments and Suggestions for Authors
The authors have satisfactorily responded to all my questions and made the necessary changes to the manuscript.
Reviewer 2 Report
Comments and Suggestions for Authors
Thank you to the authors for addressing my comments. The revised manuscript is clear and should be accepted for publication.
Comments on the Quality of English LanguageThank you to the authors for addressing my comments. The revised manuscript is clear and should be accepted for publication.